# "Why did our baby die soon after birth?"— Lessons on neonatal death in rural Cambodia from the perspective of caregivers

Ayako Suzuki[1], Mitsuaki Matsui [1,2], Rathavy Tung[1,3], Azusa Iwamoto [1,4]*

1 Project for Improving Continuum of Care with focus on Intrapartum and Neonatal Care in Cambodia, Japan International Cooperation Agency, Phnom Penh, Cambodia, 2 Department of Global Health, Nagasaki University School of Tropical Medicine and Global Health, Nagasaki, Japan, 3 National Maternal and Child Health Center, Phnom Penh, Cambodia, 4 Bureau of International Health Cooperation, National Center for Global Health and Medicine, Tokyo, Japan

* iwamoto@it.ncgm.go.jp

**Data Availability Statement:** Data cannot be disclosed according to our research protocol which was approved by ethical committees (the National Center for Global Health and Medicine in Japan

## Abstract

### Introduction

Neonatal deaths represent around half the deaths of children less than five-years old in Cambodia. The process from live birth to neonatal death has not been well described. This study aimed to identify problems in health care service which hamper the reduction of preventable neonatal deaths in rural Cambodia.

### Methods

This study adopted a method of qualitative case study design using narrative data from the verbal autopsy standard. Eighty and forty villages were randomly selected from Kampong Cham and Svay Rieng provinces, respectively. All households in the target villages were visited between January and February 2017. Family caregivers were asked to describe their experiences on births and neonatal deaths between 2015 and 2016. Information on the process from birth to death was extracted with open coding, categorized, and summarized into several groups which represent potential problems in health services.

### Results

Among a total of 4,142 children born in 2015 and 2016, 35 neonatal deaths were identified. Of these deaths, 74% occurred within one week of birth, and 57% were due to low-birth weight. Narrative data showed that three factors should be improved, 1) the unavailability of a health-care professional, 2) barriers in the referral system, and 3) lack of knowledge and skill to manage major causes of neonatal deaths.

### Conclusion

The current health system has limitations to achieve further reduction of neonatal deaths in rural Cambodia. The mere deployment of midwives at fixed service points such as health centers could not solve the problems occurring in rural communities. Community

(approval number: NCGM-G-002121-00, e-mail: kenkyushinsa@hosp.ncgm.go.jp), and the National Ethics Committee in Health Research in Cambodia (approval number: 369NECHR, website: http://www.nechr.org.kh/beta/camhrp/index.php/camhrp/index, TEL: +855-12-842-442)).

**Funding:** The study was funded by the Japan Intranational Cooperation Agency to the Project for Improving Continuum of Care with focus on Intrapartum and Neonatal Care in Cambodia; and by core funding from Nagasaki University to MM. The funders had no role in the study design, data collection and analysis, decision to publish, or preparation of the manuscript.

**Competing interests:** The authors have declared that no competing interests exist.

engagement revisiting the principle of primary health care, as well as health system transformation, is the key to the solution and potential breakthrough for the future.

## Introduction

While the total child mortality has decreased globally in the past several decades, the percentage of neonatal deaths has relatively increased [1]. From 1990 to 2018 the under-five mortality rate per 1,000 live births dropped by 59%, from 93 to 39, while neonatal mortality was reduced by 52%, from 37 to 18 [1]. Every minute five babies die in their first 28 days of life, which accounts for 46% of child deaths globally [2]. The leading causes of neonatal deaths include complications of preterm birth, birth asphyxia, and infection [3]. More than 80% of these conditions could be avoided by evidence-based and cost-effective interventions, such as prenatal corticosteroid administration for preterm births, immediate newborn care at birth, newborn resuscitation, and exclusive breastfeeding [4, 5]. However, the slower progress in the reduction of neonatal deaths indicates that appropriate measures have not been adequately implemented, especially in low- and middle-income countries.

Cambodia has achieved major successes in improving maternal, neonatal, and child health indicators over the last decade [6] and has been challenging further improvement [7, 8]. The under-five mortality rate was reduced by 72%, from 124 per 1,000 live births in 2000 to 35 in 2014. The neonatal mortality rate declined from 39 per 1,000 live births in 2000 to 18 in 2014, a slower reduction (55%) than the under-five mortality. The Ministry of Health Cambodia launched the 'Five-year Action Plan for Newborn Care in Cambodia' in 2015 [7], as a sub-sectorial document of the 'Fast Track Initiative Road Map for Reducing Maternal and Newborn Mortality (FTIRM) 2016–2020' [8]. The action plan aims to reduce neonatal mortality to 14 by the year 2020 as a milestone to the Sustainable Development Goals target. Therefore, careful monitoring of neonatal deaths is essential for assessing the effectiveness of implemented interventions.

However, an obstacle is the lack of understanding of the conditions resulting in neonatal death in communities in Cambodia. The registration of every birth and death and a review system is not yet functioning [9]. The national birth registration rate in Cambodia was estimated to be around 73% of children under five years old in 2014, while the National Strategic Plan of Identification is targeting a 90% birth registration each year by 2024 [9]. A further complication is that in Cambodia only 47% of families of the deceased receive death certificates, despite the rule that all deaths should be reported to the commune council within two weeks after the event [10].

To support the improvement of newborn health in Cambodia, the Japan International Cooperation Agency (JICA) started in May 2016 a five-year technical cooperation project, titled Improving the Continuum of Care with a Focus on Intrapartum and Neonatal Care (IINeoC). Target areas of the project are Kampong Cham and Svay Rieng provinces. In Kampong Cham, the World Health Organization supported coaching sessions on immediate newborn care from the years 2011 to 2015, while there were no previous external assistances on newborn health in Svay Rieng provinces before this study. As an initial activity, the project conducted this cross-sectional survey in both provinces to analyze the situation before implementing the planned intervention. Verbal autopsy was carried out when a neonatal death was detected during the survey, to describe the circumstances around deaths.

Verbal autopsy has been widely used to identify causes and potential contributors to deaths in countries where civil registration and vital statistics systems are under development [11].

Because of its availability, the verbal autopsy approach with written interview reports has been applied in many resource-limited settings [12]. In rural Cambodia where deaths are not reliably documented, verbal autopsy is a useful method to describe newborn death cases. The verbal autopsy questionnaire consists of several narrative sections, and using this information this study aimed to identify the problems in health care service which hamper reduction of preventable neonatal deaths in rural Cambodia.

## Methods

This study adopted a qualitative case study design using narrative data from the verbal autopsy [13, 14].

### Study sites

This study was conducted in Kampong Cham and Svay Rieng provinces in Cambodia. Both provinces are about 120 km from the capital. The socio-economic profiles of the provinces are similar, with roughly 20% of their populations living below the poverty line, slightly higher than the national average of 17% [6]. The neonatal mortality rates in 2014 were 25 and 20 per 1,000 live births in Kampong Cham and Svay Rieng, respectively, which were higher than the national average of 18 per 1,000 live births in the same year [6]. Detailed information of maternal and newborn health in the two provinces is provided in the Supporting Information S1 Table.

### Data collection

Prior to the survey, the lists of all villages were obtained from the two provincial health departments, and the selection of target villages were performed by two-stage cluster random sampling. The target villages were selected by the following procedure. First, we randomly selected eight and four communes in Kampong Cham and Svay Rieng provinces, respectively. Then, 80 and 40 villages were randomly selected from the communes for the respective provinces. The number of target villages per province was weighted according to the number of inhabitants in the two provinces, as well as the limitation of budget and duration of the survey.

Data collection was conducted in two stages. First, all households in the selected villages were visited between 16 January to 13 February 2017 to identify all births and neonatal deaths during 2015 and 2016. We set the recall period longer than the WHO recommendation (two years versus one year) because the estimated mortality cases per year were limited according to the national data. Once an infant born during this period was identified, the study team asked if the baby was alive or dead at that time, and confirmed the age at death.

As the second step, verbal autopsy using the standard record form of 'International Standard Verbal Autopsy Questionnaires Death of a Child Age Under 4 Weeks' was immediately applied when a neonatal death case was found [14]. The suspected causes of neonatal deaths were assigned using verbal autopsy information on the process of pregnancy, birth, signs, and symptoms. The surveyors referred to official documents such as maternal health record books or immunization cards if the interviewees had kept them. In cases without any records, information referring to suspected causes of death relied only on the memories of caregivers. No audio/visual recordings were used for the interview.

We employed 25 surveyors (14 females and 11 males) who had experience in conducting interviews in the latest Cambodia Demographic Health Survey (CDHS) in 2014. They received three days of training to help reconfirm and consolidate proper skills of interview with the standard questionnaires as well as to ensure the process of data collection.

## Data analysis

The narrative records of verbal autopsy were translated from the Khmer language to English, independently by two native Cambodians. When discordance in two translations was found, a third translator checked the original record and confirmed its meaning. No specific themes were defined prior to the data analysis. Information of the illness and events (Question 401) and given treatment (Question 1102) were extracted as segments and used for open-coding to identify cause of death and problems in health care services. The extracted segments within a code were compared to confirm that each code represents the same concept. Then, the codes were categorized and summarized into several groups which represent potential problems in health services, basically following the 'three delays model' [15]. We referred to the consolidated criteria for reporting qualitative research (COREQ) checklist for analyzing the narrative data and reporting the findings of the study [16]. The checklist is attached in the S1 Checklist. No specific software was used for the qualitative analysis.

Neonatal mortality rates during 2015 and 2016 for Kampong Cham and Svay Rieng provinces were computed. We obtained the estimated number of inhabitants on 1st January 2016 in all villages in the two provinces. We considered that each village is a cluster and assumed that crude birth rate in each village is same. Therefore, the estimated number of inhabitants was used as the sampling weights for adjustment in calculation of neonatal mortality rates. The statistical analysis was conducted using Stata software with the *svyset* command to adjust for the sampling weights (version 15, Stata Corp, Texas, USA).

## Ethical approval and consent

This study protocol was submitted to and received an approval from the ethics committees of the National Center for Global Health and Medicine in Japan (approval number: NCGM-G-002121-00), and the National Ethics Committee in Health Research in Cambodia (approval number: 369NECHR). Written informed consent was obtained from all participants, after explanation of the study. The participation was voluntary, and the participants could declare withdrawal from the study even after provision of the consent.

## Results

The survey team visited 23,547 households (16,792 in Kampong Cham and 6,755 in Svay Rieng) and found a total of 4,142 children (2,958 in Kampong Cham and 1,184 in Svay Rieng) born in 2015 and 2016 (Table 1). The crude birth rates during these two years were similar in the two provinces: 20 in Kampong Cham and 21 in Svay Rieng per 1,000 inhabitants, which were roughly the same as the national average of 22 [6]. We identified 35 neonatal death cases (25 in Kampong Cham and 10 in Svay Rieng) among 4,142 live births. Estimated neonatal

**Table 1. Process of identification of live births and neonatal deaths in years 2015 and 2016 in Kampong Cham and Svay Rieng provinces.**

| Variable | Kampong Cham | Svay Rieng |
|---|---|---|
| Estimated number of inhabitants | 73,439 | 28,648 |
| Number of households visited | 16,792 | 6,755 |
| Number of live births (in two years) | 2,958 | 1,184 |
| Confirmed number of neonatal deaths (in two years) | 25 | 10 |
| Neonatal mortality rate (per 1000 live births) | 9.1 | 8.2 |
| [95% CI] | [5.6–14.8] | [3.3–20.2] |

mortality rates were 9.1 [95% confidence interval 5.6–14.8] and 8.2 [3.3–20.2] per 1000 live births in Kampong Cham and in Svay Rieng, respectively.

Verbal autopsy was conducted with the caregivers of the deceased for all confirmed death cases. None of their caregivers refused or dropped out from the interviews. The respondents for the verbal autopsy were 19 mothers (53%), ten grandmothers (28%), three fathers (8%), two grandfathers (6%), one aunt, and one sibling.

## Characteristics of deceased newborn infants

The basic characteristics of all 35 neonatal death cases are shown in Table 2. Of the deaths 74% occurred within the first week after birth. Boys were dominant (60%), and 57% were low birth weight infants of less than 2.5 kg. The average age of mothers was 28 years (minimum: 19, maximum: 47), and 9% were primipara. Two women (6%) had never received antenatal care. Four babies (11%) were born by Caesarean section. Twenty-eight cases (80%) were born and 18 (51%) died at a health facility. One-third of babies received assistance to breathe. Only one family (3%) possessed an official death certificate for the baby.

**Table 2. Characteristics of neonatal deaths cases in Kampong Cham and Svay Rieng provinces, between 2015 and 2016 (n = 35).**

| Category of information | Characteristics | | n | (%) |
|---|---|---|---|---|
| Baby | Days at death | 0 | 12 | (34) |
| | | 1–6 | 14 | (40) |
| | | 7–28 | 9 | (26) |
| | Sex | Girl | 14 | (40) |
| | | Boy | 21 | (60) |
| | Birth weight | < 2.5 kg | 20 | (57) |
| | | ≥ 2.5 kg | 15 | (43) |
| Mother | Age | 15–20 | 6 | (17) |
| | | 21–30 | 17 | (49) |
| | | 31–40 | 10 | (29) |
| | | 41–50 | 2 | (6) |
| | average [min-max] | 27.5 [19–47] | | |
| | Parity | Primiparas | 3 | (9) |
| | | Multiparas | 32 | (91) |
| During pregnancy | Antenatal care | At least one contact | 33 | (94) |
| and delivery | | Never | 2 | (6) |
| | Number of fetus | Singleton | 32 | (91) |
| | | Twin | 3 | (6) |
| | Place of birth | Health facility | 28 | (80) |
| | | Outside of facility | 7 | (20) |
| | Birth attendant | Midwife/Nurse | 26 | (74) |
| | | Medical doctor | 6 | (17) |
| | | Relatives/Mother herself | 3 | (9) |
| | Mode of delivery | Vaginal | 31 | (89) |
| | | Cesarean section | 4 | (11) |
| | Neonatal resuscitation | Conducted | 12 | (34) |
| | | Not conducted | 23 | (66) |
| Death | Place of death | Health facility | 18 | (51) |
| | | Outside of facility | 17 | (49) |
| | Death certificate | Possessed | 1 | (3) |
| | | No certificate | 34 | (97) |

## Suspected causes of neonatal deaths

Table 3 summarizes the suspected causes of death from the verbal autopsy. The dominant causes of death were 'prematurity' (40%), followed by 'congenital disorders' (17%), 'asphyxia' (11%), and 'infection' (11%). In five cases we could not obtain appropriate information to estimate the cause of death. Key information used to assign the causes of death is attached in the S2 Table.

## Narratives about the process from live birth to neonatal death

From the narrative information of caregivers we categorized potential problems in health service as three parts: 1) unavailability of a health-care professional in rural communities, 2) barriers in the referral system, and 3) lack of knowledge and skill among health-care professionals to manage major causes of neonatal deaths.

**Birth and death at home due to the unavailability of a health-care professional in rural communities.**   Of seven babies who were born at home, only one was brought to a health facility after birth (Case 14), while six died at home in their first three days of life (Day 0–2). Three mothers delivered without a health-care professional. The reason why there were no assistances for their deliveries was difficulty in access to or from health facilities.

> "*Suddenly labor started at home and mother could not get to the hospital on time. The mother delivered at home without any assistance from a health staff or a traditional birth attendant*" (Case 4, Day of death: 0, Birth weight: 2.0 kg, Respondent: Grandmother)

> "*It was raining hard on the day when the baby was born. The midwife could not arrive at home on time, and the weather was very cool. Umbilical cord was cut lately after two to three hours of delivery.*" (Case 22, Day of death: 1, Birth weight: 4.3 kg, Respondent: grandmother)

**Barriers in referral resulted in neonatal deaths at home.**   There were several barriers in referral from health centers to higher level health facilities that resulted in neonatal deaths at home.

> "*The baby was premature and weak. The baby gradually became weaker until dead. The health center staff suggested to refer the baby to the provincial hospital, but no transportation was available and I did not have enough money.*" (Case 19, Day of death: 0, Birth weight: 2.0 kg, Twin pregnancy, Respondent: mother)

> "*The baby was premature. When the baby was born, the baby chocked on swallowing amniotic fluid. He was weak and had difficulty in breathing. The health staff suggested to take him to the provincial hospital. However, due to unavailability of transportation during night, the*

**Table 3. Suspected causes of neonatal death cases in Kampong Cham and Svay Rieng provinces between 2015 and 2016 (n = 35).**

| Suspected cause of death | n | (%) |
|---|---|---|
| Prematurity | 14 | (40) |
| Congenital disorders | 6 | (17) |
| Asphyxia | 4 | (11) |
| Infection | 4 | (11) |
| Respiratory distress | 1 | (3) |
| Adverse effect of vaccination | 1 | (3) |
| Unknown | 5 | (14) |

*staff then recommended to return home to warm the baby in traditional style ("Aing Phleung": roasting). There was no warming equipment at health center." (Case 27, Day of death: 0, Birth weight: 1.3 kg, Respondent: mother)*

**Deaths due to lack of knowledge and skills to manage major causes of neonatal deaths.** Twenty-eight babies were born at health facilities, and 17 died in the first three days of life. Although giving birth at a health facility is strongly recommended as a national strategy to reduce maternal and neonatal deaths, caregivers reported that staff in health facilities did not evaluate the condition of newborn infants or could not provide life-saving care to babies with complications.

"*The baby cried weakly. The size of the baby was abnormally small. The health staff at the health center suggested to put the baby to save lives into the incubator. But there was no staff who knew how to use the incubator and the baby was not kept inside it." (Case 9, Day of death: 2, Birth weight: 1.9 kg, Respondent: grandmother)*

"*The baby was premature with weakness. Soon after birth, the health staff told that the baby had already died. After the baby was born, the health staff left the baby alone. I asked her to see the dead grandson's face for the last, and noticed the baby was still breathing. I informed it then the staff started resuscitation. The baby could survive for two days and died." (Case 1, Day of death: 2, Birth weight: 1.8 kg, Respondent: grandmother)*

"*There were no emergency instruments in the health facility." (Case 25, Day of death: 0, Birth weight: 1.2 kg, Respondent: grandfather)*

## Discussion

This is a unique case study which extracted lessons on neonatal death in rural Cambodia from the perspective of caregivers. Our findings identified three problems in health care service that related to neonatal deaths: timely skilled birth attendance at delivery are not available in rural villages, there are several barriers in referral system to higher level of health facility, and quality of care in health facility is not ensured. We propose four intervention fields to solve the problems: increasing preparedness of childbirth, ensuring proximity of assistance during and immediately after delivery, support of prompt referral, and improvement of knowledge and skill in health-care professionals. Each intervention can be discussed from two viewpoints: transformation of health system and community engagement. Cross-relationships between the problems and the possible interventions are described in the S1 Fig.

To improve preparedness of childbirth, empowering community people by raising awareness on safe delivery is the first key intervention. Our study implicated that pregnant women did not aware of the signs of labor onset, which resulted in delivery at home without any assistance. If pregnant women and family members are well informed on the timing to seek necessary care, then they can take earlier actions. Integration of discussion on birth preparedness and emergency plans into antenatal care facilitates to promote timely care-seeking, since a positive relationship between high quality antenatal care and neonatal outcomes is reported [17]. The engagement of community personnel, such as village health volunteers and former traditional birth attendants can improve access and utilization of basic health services [18]. Successful experiences have been described in rural Cambodia where lay health workers played important roles in educating mothers in communities as well as accompanying women to health facilities [19].

The second possible intervention is to ensure the proximity of essential intrapartum and immediate newborn care out of health facilities, mainly at home, in collaboration with

community members. Our study found the unavailability of health-care professionals due to unfavorable environmental conditions, such as heavy weather. To overcome this issue, in India, for example, lay health workers were involved and trained in rural communities, where formal health care workers were not easily available [20, 21]. A project implemented by a non-governmental organization called SEARCH (Society for Education, Action and Research in Community Health) successfully reduced neonatal mortality by training lay health workers, who attended births with traditional birth attendants in Indian communities, and provided basic neonatal resuscitation in cases of asphyxia [22]. A typical health center in Cambodia covers around five to ten villages, and several village health volunteers including former traditional birth attendants are assigned to each village [23]. Although direct assistance for birth by lay health workers is officially prohibited in Cambodia, they may be able to provide effective support for health-care professionals to save newborn lives together. This matter requires further technical discussions and political decisions for transformation of current health system.

Enhancement of referral system is the third intervention that is an unfinished agenda in resource-limited settings. Our findings showed the unavailability of transportation hampered timely referral of newborn infants. Financial, social, and logistic supports to the referral process and coordination among different levels of health facility have accelerated the referral of patients although financial sustainability is still questionable [24]. In Cambodia, road conditions and the availability of vehicles at the primary facility level have been recently improved; however, the management of ambulances requires recurrent costs such as human resources, regular maintenance, and fuel. Covering these resources from the government budgets should be designated as a priority agenda. On the other hand, people hesitated to be referred due to fear of catastrophic expenditure in using higher level of health services. Cambodia has successfully introduced a health equity fund, which aims to reduce financial barriers for vulnerable populations [25]. However, it is apparent that transportation costs are still a barrier in the utilization of hospital services [26, 27]. Therefore, the removal of user-fees for referral services should be applied to ensure access to essential services as a part of health system transformation [28, 29].

Community engagement is also a key to make a referral system functioning. There are several successful experiences in resource-limited settings. A project in Ghana provided a low-cost emergency transportation system with a removal of user-fee, and communication tools between health facilities [30]. As the result, the utilization of referral system became higher and facility-based maternal mortality became lower in the intervention area. The project also showed that engagement of community leaders was essential to establish the referral system. Another example from Uganda showed that utilization of existing transportation systems in each community accelerated referrals of women and babies to higher level of health facility [31, 32]. A review of referral system in Burkina Faso indicated that cost sharing between the government, local stakeholders and beneficiaries was a key intervention to improve access to emergency obstetric and neonatal care [33]. These challenges tell us the importance of the fullest use of existing resources in communities, based on the principles of primary health care.

The fourth intervention focuses on how to improve knowledge and skill in health-care professionals to ensure quality care. Our findings indicated that health staff referred babies to other facilities without initial assessments and treatments. Since newborn infants experience drastic physiological changes immediately after birth, continuous monitoring of vital signs is essential as a basic procedure of initial care. However, such monitoring was not provided in the deceased cases. This would be partly caused by insufficient essential knowledge on the process of childbirth [34]. Referring to the definition of skilled birth attendant, it is the personnel who have the competency to identify and manage or refer women and/or newborn infants with complications [35]. In this regard our study indicates that some health-care professionals

are unskilled, and therefore systematic efforts are required to make them competent. Previous studies showed that knowledge and skills were deteriorated if active continuous education had not been provided [36, 37]. Daily clinical services do not provide sufficient opportunities to learn how to manage severe complications in newborn infants, especially for those who work in rural health centers, because the number of births is limited. To overcome this situation, regular training using simulation with periodic supportive supervisions is a promising tool to enhance competencies of health care providers as well as to improve quality of care [38, 39].

A major limitation of this study was the retrospective nature of verbal autopsy, especially in recall bias among the respondents. Information on the process from the birth to death relied on family member memories up to two years old. Such blurred memories may cause assumptions on the part of the surveyors, which were recorded as the causes of deaths. It is important to tell the interviewers to dictate their narratives accurately without modification through their trainings before the survey.

This recall bias may also explain the lower neonatal mortality rates than those in CDHS 2014 (9.1 vs 25 in Kampong Cham, 8.2 vs 20 in Svay Rieng). It is known that the survey estimates are sometimes biased to underestimate the mortality [40]. Neonatal deaths in this study could be under-reported because the caregivers did not disclose the events of birth and death, or the exact dates of birth and death were not identified. However, considering the recent improvement of the health status and economic situation in Cambodia, our estimates would reflect the reality of neonatal deaths since the CDHS 2014 results estimated neonatal mortality for several years preceding the survey. Although we do not have the means to verify the exact neonatal mortality in this study, it did not affect the qualitative results regarding the caregivers' perspectives. Establishment of an effective vital registration system with perinatal death review, for all births and deaths, will provide more exact figures on neonatal deaths, although it is a challenge in limited resource settings [41–43].

This study showed that the current health system has limitations to achieve further reduction of neonatal deaths in rural Cambodia. The government has successfully improved the status of maternal and neonatal health by systematic deployment of midwives in health centers, facilitating deliveries at health facilities, and making tiers of health centers and referral hospitals [44]. However, the mere deployment of midwives at fixed service points such as health centers could not solve the problems happening in rural communities. Community engagement revisiting the principle of primary health care [45], as well as health system transformation, is the key to the solution and potential breakthrough for the future.

## Supporting information

**S1 Checklist. Consolidated criteria for reporting qualitative studies (COREQ): 32-item checklist.**
(PDF)

**S1 Fig. Relationships between the findings and the possible interventions from this study.**
(PDF)

**S1 Table. Summary of the characteristics on demography and health service in Kampong Cham and Svay Rieng provinces, and Cambodia.**
(PDF)

**S2 Table. Summary of the process from birth to death and suspected causes in 36 neonatal deaths cases in Kampong Cham and Svay Rieng provinces, between 2015 and 2016.**
(PDF)

## Acknowledgments

We express our gratitude to Professor Emeritus Vincent De Brouwere for his technical inputs. We also appreciate Dr Masahiko Hachiya for his technical advice. We also acknowledge the respondents of this survey, without whom the study could not have been conducted.

## Author Contributions

**Conceptualization:** Mitsuaki Matsui, Rathavy Tung, Azusa Iwamoto.

**Data curation:** Mitsuaki Matsui, Azusa Iwamoto.

**Formal analysis:** Ayako Suzuki, Mitsuaki Matsui, Rathavy Tung, Azusa Iwamoto.

**Funding acquisition:** Rathavy Tung, Azusa Iwamoto.

**Investigation:** Ayako Suzuki, Mitsuaki Matsui, Azusa Iwamoto.

**Methodology:** Ayako Suzuki, Mitsuaki Matsui, Rathavy Tung, Azusa Iwamoto.

**Project administration:** Rathavy Tung, Azusa Iwamoto.

**Resources:** Azusa Iwamoto.

**Software:** Mitsuaki Matsui.

**Supervision:** Rathavy Tung, Azusa Iwamoto.

**Validation:** Ayako Suzuki, Mitsuaki Matsui, Rathavy Tung, Azusa Iwamoto.

**Visualization:** Ayako Suzuki, Mitsuaki Matsui.

**Writing – original draft:** Ayako Suzuki.

**Writing – review & editing:** Mitsuaki Matsui, Azusa Iwamoto.

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
