## [Decision Letter · Decision Letter 0]

11 Nov 2020

PONE-D-20-32314

"Why did our baby die soon after birth?" – lessons on neonatal death in rural Cambodia from the perspective of caregivers.

PLOS ONE

Dear Dr. Azusa Iwamoto,

Thank you for submitting your manuscript to PLOS ONE. After careful consideration, we feel that it has merit but does not fully meet PLOS ONE’s publication criteria as it currently stands. Therefore, we invite you to submit a revised version of the manuscript that addresses the points raised during the review process.

We look forward to receiving your revised manuscript.

Kind regards,

Masamine Jimba

Academic Editor

PLOS ONE

Journal Requirements:

2.) Please provide details of how the data was analysed in the Methods section.

3.) We note that you have indicated that data from this study are available upon request. PLOS only allows data to be available upon request if there are legal or ethical restrictions on sharing data publicly. For more information on unacceptable data access restrictions, please see http://journals.plos.org/plosone/s/data-availability#loc-unacceptable-data-access-restrictions.

4.) PLOS requires an ORCID iD for the corresponding author in Editorial Manager on papers submitted after December 6th, 2016. Please ensure that you have an ORCID iD and that it is validated in Editorial Manager. To do this, go to ‘Update my Information’ (in the upper left-hand corner of the main menu), and click on the Fetch/Validate link next to the ORCID field. This will take you to the ORCID site and allow you to create a new iD or authenticate a pre-existing iD in Editorial Manager. Please see the following video for instructions on linking an ORCID iD to your Editorial Manager account: https://www.youtube.com/watch?v=_xcclfuvtxQ

5.) Please include captions for your Supporting Information files at the end of your manuscript, and update any in-text citations to match accordingly. Please see our Supporting Information guidelines for more information: http://journals.plos.org/plosone/s/supporting-information.

Additional Editor Comments (if provided):

This is an interesting article in the field of maternal and child health in a lower-middle income country. Its contents will be improved by responding to the specific comments from two reviewers. The following points should be also considered to improve its quality.

L20: The abstract should be revised after improving the main text.

L24: This study aimed to…(the same for the main text)

L73: The objective statement is not strong enough. “Drawing lessons” are too general. It should be more specific.

L79: This study seems to be conducted by using a mixed-methods research design, but it was not considered as such. If not, please explain it. The authors’ may read several articles about the mixed methods research, for example, by Creswell and reconstruct the methods section. For qualitative research part, it is better to attach COREQ check list. It will strengthen the credibility of the narrative records. It is also better to write how “validity threats” were overcome. (See Maxwell JA. Qualitative research design: an interactive approach. 3rd ed. SAGE, 2013.) The method of data analysis is not clear, either.

L 139: For each %, % is not necessary. Just 34 is enough. The same for Table 2.

L196: In many paragraphs, the first sentences started with general statement, and then it was supported by the authors’ findings. This style should be changed, not from general to specific, but from specific to general. Please show the finding of this study first and then discuss it using references and other findings of this study. One paragraph is too long to explain each topic sentence.

L 198, 199: This sentence is not necessary. Please make a summary of major findings. “Reveal” is a very strong verb, which should be avoided in this study design.

L203: It looks like “causality” was proven about the unavailability of SBAs in this study.

L257: Please add one topic sentence before starting the paragraph.

L310: The Royal Government…

Fig 1: It is not clear.

Reviewers' comments:

Reviewer's Responses to Questions

**Comments to the Author**

1. Is the manuscript technically sound, and do the data support the conclusions?

Reviewer #1: Yes

Reviewer #2: Yes

2. Has the statistical analysis been performed appropriately and rigorously? 

Reviewer #1: No

Reviewer #2: N/A

3. Have the authors made all data underlying the findings in their manuscript fully available?

Reviewer #1: Yes

Reviewer #2: No

4. Is the manuscript presented in an intelligible fashion and written in standard English?

Reviewer #1: Yes

Reviewer #2: Yes

5. Review Comments to the Author

Reviewer #1: This study analyzed the results of verbal autopsy for neonatal mortality cases in two rural provinces in Cambodia. It highlights important barriers that might affect neonatal death in a country that is often regarded as a success case for improving maternal, newborn, and child health. I believe that the insights presented in the manuscript should be widely shared. I hope the points listed below would contribute to improving the manuscript, if relevant.

1. The authors stated that “the reduction of neonatal deaths has stagnated.” And they presented data as “From 1990 to 2018 the under-five mortality rate per 1,000 live births dropped by 59%, from 93 to 39, while neonatal mortality was reduced by 52%, from 37 to 18 [1].” (Line 2-) This shows that the percentage of reduction is 59% for under-five mortality and 52% for neonatal mortality. It sounds a bit too strong to state that “the reduction of neonatal deaths has stagnated” compared to child mortality.

2. In the Introduction section, the authors can explain how verbal autopsy has been implemented in resource-limited settings, by referring to previous studies that tackled its possible methodological issues. These issues may include how to improve the identification of possible causes of deaths, particularly when death certificates are not available and when the recall period is longer than the recommended period? The authors may notice that WHO recommended that the recall period does not exceed one year of death (for example, https://www.who.int/healthinfo/statistics/WHO_VA_2012_RC1_Instrument.pdf?ua=1). The authors may relate these issues to be mentioned in the Background section to the Methods section (how the data collection minimizes potential errors) and the Discussion section (strengths and limitations of this study).

3. The authors may explain more about the “Study sites” subsection (Line 80-). Were there any activities (other than verbal autopsy) in the JICA project, which could potentially reduce neonatal mortality? It is unclear if the verbal autopsy in this study could be interpreted as the one without project intervention or not.

4. Related to the point above, were there any ongoing or previous external assistance (by JICA and other organizations) in the study sites related to neonatal health? This information may be important to interpret neonatal mortality rate estimates and the cause of deaths in this study.

5. Related to the point above, the “Study sites” subsection may need more about the level of maternal, newborn, and child health services in the study sites, in addition to neonatal mortality rates. Such information may include the followings: the coverage of facility-based delivery, delivery assisted by skilled birth attendants, and the mean number of antenatal care received. In addition, the authors may need to explain the hierarchical structure of health facilities, the availability of midwives and delivery assistance in different tiers of health facilities, the availability of transportation means owned by a health facility (such as ambulance), and the activities of village-level health volunteers, if any, in the study site. Such information may help readers to interpret the causes of deaths presented in this study and understand the Discussion section.

6. How was the training of survey team members conducted? Is there any supervision or auditing of survey team members to improve the quality of the cause of death identification? They may be related to how this study minimized biases in the cause of death identification.

7. How were neonatal death cases distinguished from stillbirth cases? The questionnaire used in this study may be able to distinguish between them, if it is properly used. The authors could explain how to minimize potential errors.

8. Reference number 12 ('International Standard Verbal Autopsy Questionnaires Death of a Child Age Under 4 Weeks') is the 2007 version. Are there reasons why the authors did not use the 2012 version or later?

9. In Lines 101-, the authors explained how the suspected causes of deaths were documented. In the data collection, how were the data items in the Supplementary Table (such as the birth weight, gestational months, and diagnosis in hospitals) identified? Did survey team members refer to any documents that interviewees held or a health facility possessed? What if these documents were unavailable? The authors could explain more about data collection process and validation.

10. In Lines 118-, the authors explained that “neonatal mortality rates were also similar in the two provinces at approximately 8 per 1,000 live births.” Did this estimate reflect the complex survey design the authors used (two-stage cluster sampling)? The authors concluded that the estimated neonatal mortality rates were lower than the provincial estimates, according to DHS. However, the authors may be able to estimate the confidence interval and interpret if the estimate is different from the DHS estimate even after considering the confidence interval.

11. Related to the point above, the authors could note that DHS estimates of neonatal mortality were possibly influenced by the undercount of mortality cases and thus corrected by multiple methods (for example, https://www.un.org/en/development/desa/population/publications/pdf/technical/TP2011-2_MortEstMajorSampSurv.pdf). That is, neonatal mortality rate based on DHS may be corrected its undercounts. Therefore, the authors’ neonatal counts might not necessarily be worse than counts done by DHS. The authors may be able to justify their methods by pointing out that such undercounts might be inevitable.

12. The authors analyzed narratives by focusing on services related to childbirth (such as availability of SBAs, knowledge and skills related to neonatal deaths, and referral) (Lines 153-). I totally agree that these factors are important to reduce neonatal deaths. In addition, were there issues related to services in the pregnancy stage identified, such as early detection and management of maternal complications and advice related to birth preparedness?

13. The authors proposed the training of lay village workers dispatched in rural communities to improve access to SBAs (Line 206-). Similar efforts might have already been started in, at least, some part of Cambodia (for example, https://resourcecentre.savethechildren.net/node/16699/pdf/Learning%20package%202%20-%20Role%20of%20volunteers%20FINAL_low%20res%20for%20WEB.pdf;
http://unicefcambodia.blogspot.com/2015/04/volunteers-crucial-to-health-delivery.html). The authors may be able to explain how it could work in the study site, by referring to past efforts in Cambodia.

14. The authors may be able to document their lesson learnt for those who conduct verbal autopsy, by focusing on how verbal autopsy can be improved at resource-limited settings.

15. I believe the data has been made available in Supplementary Table, on the contrary to the data availability statement (“some restriction may apply”).

Reviewer #2: [Abstract]

Please add the total number of births identified in the study sites.

[Introduction]

Please mention the birth registration rate in Cambodia. (Line 60)

Please provide common reasons for underreport of death in Cambodia.

Please provide general information on maternal newborn and child health care program in Cambodia, including healthcare system, types of health care providers, and health insurance for health care services.

How many days do mothers and babies stay at facility after giving birth?

Please replace some information about study design and setting (Lines 63-72) to the Method section.

[Methods]

In the study site section, please provide the population of the two provinces.

Please describe more clearly about the second stage of sampling because I wondered if authors randomly selected villages per a commune after determining the number of villages to be selected per commune in advance, or randomly selected 80 villages from 8 communes in Kampong Cham and 40 villages from 4 communes in Svay Rieng regardless of commune size. (Lines 88-94)

Please provide the reference for the analysis of verbal autopsy. (Line 104)

Please explain more detail about how the narrative data on the process from live birth to neonatal death was analyzed. (Lines 104-106)

Verbal autopsy potentially gives emotional distress to respondents. How did the study consider for it?

[Results]

Please show the number of neonatal death cases first, followed by neonatal mortality rates. (Lines 118-121)

[Discussion]

Please discuss the results of neonatal mortality rates and suspected causes of death, although the authors have explained potential reasons for the lower neonatal mortality rate observed in the study compared to the national representative data.

6. PLOS authors have the option to publish the peer review history of their article (what does this mean?). If published, this will include your full peer review and any attached files.

Reviewer #1: No

Reviewer #2: No

---

## [Author Response · Author response to Decision Letter 0]

5 Jan 2021

No specific responses to specific editor and reviewers.

---

## [Decision Letter · Decision Letter 1]

3 Feb 2021

PONE-D-20-32314R1

"Why did our baby die soon after birth?" – lessons on neonatal death in rural Cambodia from the perspective of caregivers.

PLOS ONE

Dear Dr. Iwamoto,

Thank you for submitting your manuscript to PLOS ONE. After careful consideration, we feel that it has merit but does not fully meet PLOS ONE’s publication criteria as it currently stands. Therefore, we invite you to submit a revised version of the manuscript that addresses the points raised during the review process.

We look forward to receiving your revised manuscript.

Kind regards,

Masamine Jimba

Academic Editor

PLOS ONE

Additional Editor Comments (if provided):

Now the research design is clear. To improve the quality of the manuscript, please try to improve the following points

1. P2L1: In the abstract, please add subheadings like other PLoS One articles: Introduction, Methods, Results, and Conclusion.

2. P2L3: …neonatal deaths have not been well described…

3. P2L5: aimed to…(In this type of research, the objective statement should be written in the past tense.)

4. P2L6: This study adopted a qualitative case study design using…

5. P2L12: Please add a data analysis method.

6. P2L14: Narrative data showed that the following three (themes?factors?) should be improved……mortality. They were insufficient availability…….., and barriers in referral.

7. P2L17: …could be also effective to facilitate…..residents for reducing neonatal mortality…

8. P4L65: This study aimed to…

9. P5L68: This study adopted a qualitative case study design using…

10. P5L78: This comment is optional. In this section two writing styles are used and mixed up: “We…” or using “passive tense” without using “we.” This affects readability and it is better to stick to one style. Please decide whether to keep as it is, or to make the style consistent.

11. P5L71: Prior to qualitative data collection, a survey was conducted in…

12. P5L75: …which were higher than the national average….live births in the same year (OK?)

13. P5L81: …, which aimed to reconfirm…

14. P6L103: Data analysis method is not clearly stated in here. Thematic analysis, grounded theory, or any other method was used for data analysis?

15. P12L194: There were…

16. P12L211: It is a risky to use the term “causality” even if “potential” is added. You may simply say, “The first challenge is the unavailability of SBAs due to….”

17. P12L213: This is a recommendation statement, which is usually not appropriate to do it in the middle of discussion. All the recommendations should come before or after the final conclusion. This sentence can be rewritten as follows, for example. “To realize delivery by an SBA, two approach has been taken. The first one is to improve it from the service supply side. In India, for example, lay village health workers were involved and trained in rural communities, where SBAs were not easily available (16,17).

18. P13L218…had reduced neonatal… (Please use “verbs” instead of “noun +of”. Overusing nouns makes a sentence unreadable.)

19. P13L224: …emergency. In Stung Trend province in Cambodia, the key to successful community involvement depends on…

20. P13L227: The second approach for improving delivery by SBAs is strengthening (or empowering?) the demand side. By this, community people can raise awareness about safe delivery. The neonatal…

21. P14L242: It sounds like you are discussion ref. 24, and your finding is used as a supporting evidence for the ref 24. Although it is not supporting the message of ref. 24, the key message in this paragraph seems to be ref. 24 and not yours. This paragraph is too long and redundant.

22. P15L261: Again, you are discussing ref. 21 and 30. Please show your finding first and discuss it, not someone’s. On L209, “enhancement of referral systems” is used as one of major findings, this term should appear in the first sentence, not in the last one.

23. P16L287: Again you are discussing ref 34, 35, and 37, rather than discussing your unique findings. It sounds like you are repeating “Introduction” in here and because of this, this last paragraph is very weak to show the strength of this study. Despite several limitations mentioned in the manuscript, what you can say with confidence?

Reviewers' comments:

Reviewer's Responses to Questions

**Comments to the Author**

1. If the authors have adequately addressed your comments raised in a previous round of review and you feel that this manuscript is now acceptable for publication, you may indicate that here to bypass the “Comments to the Author” section, enter your conflict of interest statement in the “Confidential to Editor” section, and submit your "Accept" recommendation.

Reviewer #1: All comments have been addressed

Reviewer #2: All comments have been addressed

2. Is the manuscript technically sound, and do the data support the conclusions?

Reviewer #1: Yes

Reviewer #2: Partly

3. Has the statistical analysis been performed appropriately and rigorously? 

Reviewer #1: Yes

Reviewer #2: Yes

4. Have the authors made all data underlying the findings in their manuscript fully available?

Reviewer #1: No

Reviewer #2: Yes

5. Is the manuscript presented in an intelligible fashion and written in standard English?

Reviewer #1: Yes

Reviewer #2: Yes

6. Review Comments to the Author

Reviewer #1: The authors added more explanations on the study site, the method of a verbal autopsy, data collection and analysis, interpretations, and potential biases. I have several comments on the revised part of the manuscript.

1. Regarding the added explanations on neonatal mortality estimates in the Methods section, the authors may need to explain the methods of estimating the confidence interval by clarifying a particular estimation method (for example, Rao-Scott Chi-square test) or particular Stata command.

2. In contrast to the revised objective statement (…to identify the problems in health care service), the Data collection subsection in the Methods section did not specifically mention how such problems in health care service were collected (the authors explained how mortality cases were identified and how the process of pregnancy, birth, signs, and symptoms were identified, though). The authors may clarify in the Methods section regarding how 'International Standard Verbal Autopsy Questionnaires Death of a Child Age Under 4 Weeks' were used to identify problems in health care service.

3. As the authors characterize this study as a qualitative study in the revised manuscript, they may want to explain the methods according to the standard guidelines, not simply added the checklist. Regarded “N/A” items in the added COREQ checklist, the authors may be able to clarify the following items in the main text and the checklist: #2-#4 (background of 40 survey team members), #13 (whether there were non-participations among identified cases), #19 (clarify if audio/visual recording was not used), #20 (“International Standard Verbal Autopsy Questionnaires Death of a Child Age Under 4 Weeks” were used, so can be clarified in the checklist), #21 (typical time duration for the interview), #23 (clarify no if not returned), and #24-25 (coding strategy).

4. Related to the comment above, the authors may be able to clarify how to extract themes from the verbal autopsy record. The authors identified “unavailability of a SBA,” “lack of knowledge and skills to manage major causes of neonatal deaths,” and “Barriers in referral.” How did the authors finally achieve to confirm these themes?

Reviewer #2: Thank you for addressing all my comments. I have a few more comments as follows.

I suggest to add the authors' responses in the main text: #5 the national recommendation on postpartum care, and #11 ethical considerations for respondents during verbal autopsy.

Thank you for creating S2 Table. This is very informative. I suggest the authors to add simple descriptions or key words regarding the process from live birth to neonatal death for each case.

I agree that the discussion in the last paragraph is important. However, I do not really see it the conclusion of the manuscript. Please provide a clearer conclusion.

7. PLOS authors have the option to publish the peer review history of their article (what does this mean?). If published, this will include your full peer review and any attached files.

Reviewer #1: No

Reviewer #2: No

---

## [Author Response · Author response to Decision Letter 1]

15 Mar 2021

Our responses to the Academic Editor, Reviewer 1, and Reviewer 2 are described in the table of ‘Response to Reviewers’.

---

## [Decision Letter · Decision Letter 2]

28 Mar 2021

PONE-D-20-32314R2

"Why did our baby die soon after birth?" – lessons on neonatal death in rural Cambodia from the perspective of caregivers.

PLOS ONE

Dear Dr. Iwamoto,

Thank you for submitting your manuscript to PLOS ONE. After careful consideration, we feel that it has merit but does not fully meet PLOS ONE’s publication criteria as it currently stands. Therefore, we invite you to submit a revised version of the manuscript that addresses the points raised during the review process.

We look forward to receiving your revised manuscript.

Kind regards,

Masamine Jimba

Academic Editor

PLOS ONE

Journal Requirements:

Additional Editor Comments (if provided):

The revised manuscript was greatly improved. In addition to two reviewers' additional minor comments, please try to improve the following points.

1. Abstract: The verb 'reveal' is often used in more rigor study design articles. It is better to use 'show' or other weaker verbs. As two reviewers are suggesting to modify the conclusion of the main text, the conclusion of the abstract also may be slightly changed.

2. P15 L273: "unfortunately" can be cut.

Reviewers' comments:

Reviewer's Responses to Questions

**Comments to the Author**

1. If the authors have adequately addressed your comments raised in a previous round of review and you feel that this manuscript is now acceptable for publication, you may indicate that here to bypass the “Comments to the Author” section, enter your conflict of interest statement in the “Confidential to Editor” section, and submit your "Accept" recommendation.

Reviewer #1: All comments have been addressed

Reviewer #2: (No Response)

2. Is the manuscript technically sound, and do the data support the conclusions?

Reviewer #1: Yes

Reviewer #2: Partly

3. Has the statistical analysis been performed appropriately and rigorously? 

Reviewer #1: Yes

Reviewer #2: Yes

4. Have the authors made all data underlying the findings in their manuscript fully available?

Reviewer #1: Yes

Reviewer #2: Yes

5. Is the manuscript presented in an intelligible fashion and written in standard English?

Reviewer #1: Yes

Reviewer #2: Yes

6. Review Comments to the Author

Reviewer #1: Thank you very much for considering the comments made by reviewers provided in the previous round or the peer-review process. I have minor comments as follows:

1. In Lines 130-, the authors stated that “We considered that each village is a cluster and assumed that similar crude birth rates exist for each village.” However, it may not be a reason for the adjustment using the sampling weight. This study employed cluster sampling, under which it selected 80 and 40 villages first, then enumerated all the birth and neonatal death cases in the selected villages. Suppose the authors employed the simple random sampling method (without considering the probability-proportionate-to-sizes of villages). In that case, each case has a different probability of being selected in the sample due to differences in villages' population sizes. This point may be a reason for the adjustment for sampling weight in the estimate of neonatal mortality.

2. The authors modified the Discussion section to highlight the importance of ensuring quality care, enhancing the referral system. Although they also modified the concluding statement (Lines 316-) to emphasize the importance of seeking solutions at the community level, this statement seems not reflect the issues of quality care and referral system. Since the Discussion section was reorganized based on the three-delay model, the concluding statement may also be structured according to the model.

Reviewer #2: I confirmed that the authors have addressed my previous suggestions.

I have some minor comments as follows.

1. Page17 Line320 I did not understand what “at fixed points” means. Please rephrase it.

2. Page17 Line316-325. I acknowledge that the authors know the real situation in rural Cambodia very well and provided very important recommendations at the end of the paper. However, as a research paper, the conclusion can be described in more simple and specific manners, based on the study findings.

3. Please add footnotes for S3 Table, and describe why information on cases 1, 4, 9, 19, 22, 25, and 27 is written by Bold Italic characters.

7. PLOS authors have the option to publish the peer review history of their article (what does this mean?). If published, this will include your full peer review and any attached files.

Reviewer #1: No

Reviewer #2: No

---

## [Author Response · Author response to Decision Letter 2]

24 Apr 2021

Our responses to the editor and to the reviewers are mentioned in another file ('Response to Reviewers').

---

## [Decision Letter · Decision Letter 3]

5 May 2021

PONE-D-20-32314R3

"Why did our baby die soon after birth?" – lessons on neonatal death in rural Cambodia from the perspective of caregivers.

PLOS ONE

Dear Dr. Azusa Iwamoto,

Thank you for submitting your manuscript to PLOS ONE. After careful consideration, we feel that it has merit but does not fully meet PLOS ONE’s publication criteria as it currently stands. Therefore, we invite you to submit a revised version of the manuscript that addresses the points raised during the review process.

We look forward to receiving your revised manuscript.

Kind regards,

Masamine Jimba

Academic Editor

PLOS ONE

Journal Requirements:

Additional Editor Comments (if provided):

In this revision, the manuscript was greatly improved, but it needs further revision in the discussion.

1. P15L244: The paragraph is well written but the authors do not show any data (evidence) of this study. As I mentioned previously, this is a place where the authors should discuss their own unique data, but they are missing here.

2. P15L254: The same as above. No data are shown from this study.

3. P16L268: This paragraph is good as the authors are discussing their own data.

4. P17L295: This topic sentence is not clear. In the previous paragraphs, the specific contents of each intervention are clearly written, but this sentence is just focusing on how to ensure quality care and it does not seem to be a specific intervention. In P15L241, the fourth intervention is described as "development of human resources," but this phrase is missing here.

Reviewers' comments:

Reviewer's Responses to Questions

**Comments to the Author**

1. If the authors have adequately addressed your comments raised in a previous round of review and you feel that this manuscript is now acceptable for publication, you may indicate that here to bypass the “Comments to the Author” section, enter your conflict of interest statement in the “Confidential to Editor” section, and submit your "Accept" recommendation.

Reviewer #1: All comments have been addressed

Reviewer #2: All comments have been addressed

2. Is the manuscript technically sound, and do the data support the conclusions?

Reviewer #1: Yes

Reviewer #2: Yes

3. Has the statistical analysis been performed appropriately and rigorously? 

Reviewer #1: Yes

Reviewer #2: Yes

4. Have the authors made all data underlying the findings in their manuscript fully available?

Reviewer #1: No

Reviewer #2: Yes

5. Is the manuscript presented in an intelligible fashion and written in standard English?

Reviewer #1: Yes

Reviewer #2: Yes

6. Review Comments to the Author

Reviewer #1: All the comments made in the previous round of the review process have been addressed in the current manuscript.

Reviewer #2: I confirmed that the authors have addressed all of my suggestions, and thank them for their hard work since the first submission. Congratulations. I believe that this manuscript will save more newborns in rural Cambodia.

7. PLOS authors have the option to publish the peer review history of their article (what does this mean?). If published, this will include your full peer review and any attached files.

Reviewer #1: No

Reviewer #2: No

---

## [Author Response · Author response to Decision Letter 3]

18 May 2021

Our responses to the Academic Editor are described in the table of ‘Response to Reviewers’. All revision parts were underlined in the ‘Revised Manuscript with Track Changes’. To show the relationships between our findings and the possible interventions from this study, we add the Supporting information S4 Figure.

---

## [Editor Report · Decision Letter 4]

20 May 2021

"Why did our baby die soon after birth?" – lessons on neonatal death in rural Cambodia from the perspective of caregivers.

PONE-D-20-32314R4

Dear Dr. Azusa IWAMOTO,

We’re pleased to inform you that your manuscript has been judged scientifically suitable for publication and will be formally accepted for publication once it meets all outstanding technical requirements.

Kind regards,

Masamine Jimba

Academic Editor

PLOS ONE

Additional Editor Comments (optional):

Thank you very much for your hard work to revise this important article for many times. Now it is ready for publication. Congratulations! This is optional, but you may consider to remove "This study showed that" in the conclusion of the abstract. Staring with "The current health system..." seems to make sense. You may make a decision at proof-correction stage.
---

## [Editor Report · Acceptance letter]

24 May 2021

PONE-D-20-32314R4 

"Why did our baby die soon after birth?" – lessons on neonatal death in rural Cambodia from the perspective of caregivers. 

Dear Dr. Iwamoto:

I'm pleased to inform you that your manuscript has been deemed suitable for publication in PLOS ONE. Congratulations! Your manuscript is now with our production department. 

Kind regards, 

on behalf of

Professor Masamine Jimba 

Academic Editor

PLOS ONE